# Titanium catalyzed [2σ + 2π] cycloaddition of bicyclo[1.1.0]-butanes with 1,3-dienes for efficient synthesis of stilbene bioisosteres

Yonghong Liu[1,2,5], Zhixian Wu[1,2,5], Jing-Ran Shan[3] ✉, Huaipu Yan[2], Er-Jun Hao[4] ✉ & Lei Shi [1,2,4] ✉

Natural stilbenes have shown significant potential in the prevention and treatment of diseases due to their diverse pharmacological activities. Here we present a mild and effective Ti-catalyzed intermolecular radical-relay [2σ + 2π] cycloaddition of bicyclo[1.1.0]-butanes and 1,3-dienes. This transformation enables the synthesis of bicyclo[2.1.1]hexane (BCH) scaffolds containing aryl vinyl groups with excellent regio- and *trans*-selectivity and broad functional group tolerance, thus offering rapid access to structurally diverse stilbene bioisosteres.

Benzene rings are crucial chemical structural components that are commonly found in pharmaceutical molecules and natural products[1–4]. However, in physiological conditions, drug molecules with multiple phenyl rings often undergo metabolic modifications that compromise their solubility and metabolic stability[5]. Therefore, enhancing key pharmacokinetic properties to broaden the range of potential drug molecules is crucial in clinical trials[6–11]. Scientists have discovered that by replacing a benzene group with different sp³-rich hydrocarbon bioisosteres[12–16], such as bicyclo[1.1.1]pentane (BCP)[17–21], bicyclo[2.1.1] hexane (BCH)[22,23], bicyclo[3.1.1]heptane (BCHep)[24–27], bicyclo[2.2.2] octane[28,29], and cubane[30,31], the properties of drugs can be significantly enhanced. For example, BCP-avagacestat[32] and BCH-valsartan[33] showed improved water solubility compared with the original drugs (Fig. 1a).

Natural stilbenes are an important class of polyphenolic phytochemicals characterized by two phenyl groups connected by a vinyl bridge[34–37]. Pharmacological research has demonstrated that stilbenes have significant potential in the prevention and treatment of diseases due to their anticancer, anti-inflammatory, neuroprotective, antibacterial, and antifungal properties[38–41]. Resveratrol, a well-known stilbene widely used in Chinese and Japanese folk medicine, has shown promise[42–45]. Nevertheless, its poor bioavailability has hindered its clinical progress. An intriguing strategy to significantly enhance the pharmacokinetic properties of resveratrol involves substituting the phenol ring with a BCP moiety (Fig. 1b)[46]. However, the synthesis requires 9-step reactions from a BCP-acid derivative. Given their wide

range of biological activities and immense potential for diseases treatment, it is imperative to develop a general and efficient synthetic approach for exploring the bioisosteres of stilbenes.

The BCH scaffolds, which can imitate both *ortho-* and *meta-*substituted benzenoids, have gained significant interest. In addition to the intramolecular cycloaddition of 1,5-dienes[23,47,48], the strain-release-driven [2σ + 2π] cycloaddition of bicyclo[1.1.0]-butanes (BCBs) with a single alkene becomes a popular approach for efficiently synthesizing BCH scaffolds. Very recently, significant advancements were made independently by Glorius[22,49–51] and Brown[52] in demonstrating the intermolecular [2π + 2σ]-photocycloaddition through triplet energy transfer. Procter disclosed elegant SmI₂-catalysed insertion of electron-deficient alkenes into BCB ketones via radical relay mechanism[53]. Li[54] and Wang[55] reported the boryl radical-initiated cycloaddition of BCBs with simple alkenes. Additionally, Lewis acid catalyzed formal (3 + 2)-cycloaddition leads to the formation of diverse heteroatom-substituted BCH scaffolds[56–59].

Among the various transition metals, titanium stands out as one of the most affordable options that are generally considered nontoxic and environmentally friendly[58]. The combination of these desirable characteristics, alongside its rich redox properties, makes titanium particularly appealing for medicinal chemistry[60,61].

In this work, we present a mild and effective Ti-catalyzed intermolecular radical [2σ + 2π] cycloaddition of bicyclo[1.1.0]-butanes and 1,3-dienes (Fig. 1c). This transformation enables the synthesis of BCH

[1]Cancer Hospital of Dalian University of Technology, 116024 Dalian, China. [2]School of Chemistry, Dalian University of Technology, 116024 Dalian, China. [3]Department of Chemistry and Biochemistry, University of California Los Angeles, Los Angeles, CA 90095, USA. [4]Key Laboratory of Green Chemical Media and Reactions, Ministry of Education, School of Chemistry and Chemical Engineering, Henan Normal University, Xinxiang 453007, China. [5]These authors contributed equally: Yonghong Liu, Zhixian Wu. ✉e-mail: jrshan@chem.ucla.edu; hej@htu.edu.cn; shilei17@dlut.edu.cn

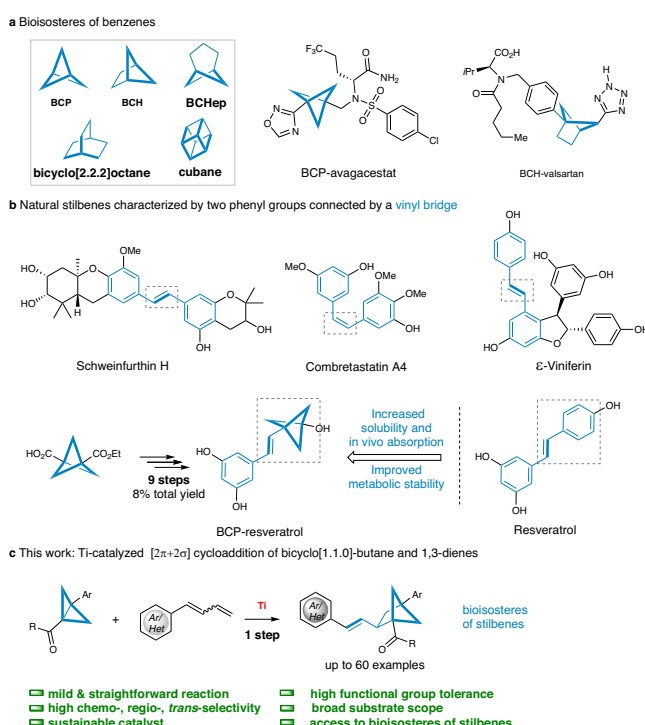

**Fig. 1 | Approaches for the synthesis bioisosteres of benzenes and stilbenes.** **a** Bioisosteres of benzenes; **b** representative natural stilbenes; **c** Ti-catalyzed [2σ + 2π] cycloaddition to obtain stilbene bioisosteres.

scaffolds containing aryl vinyl groups with excellent regio- and *trans*-selectivity and broad functional group tolerence, thus offering rapid access to structurally diverse stilbene bioisosteres.

## Results

Optimization of reaction conditions. The study began by using phenyl(3-phenylbicyclo[1.1.0]butan-1-yl)methanone **1a** and a mixture of *E/Z* 1-phenyl-1,3-butadiene **2a** as coupling partners. After making preliminary optimizations to the conditions, the reaction was conducted with the presence of Cp$_2$TiCl$_2$, Mn powder, and Et$_3$N•HCl in 1,4-dioxane for a duration of 48 h at room temperature. Excitingly, this resulted in the desired *trans*-configuration product **3a** with a moderate yield of 61% (Table 1, entry 1). The structure of **3a** was unambiguously confirmed through X-ray analysis (for more details see the Supplementary Fig. 14). When CpTiCl$_3$ was used instead of Cp$_2$TiCl$_2$, only 19% of the product **3a** was obtained (entry 2). In order to further improve reaction efficiency, we have focused on the selection of salen-Ti as the catalyst. 1,2-diphenylethylenediamine-based salen-Ti **I** has been successfully used by Lin in the [3 + 2] cycloaddition of cyclopropyl ketones and alkenes[62,63], however, this catalyst did not effectively increase the yield of product **3a** (entry 3). Notably, the diamine backbone of the salen ligand was found to significantly influence the reaction and proved critical in achieving a high yield of the product. For instance, altering the diamine backbone to a 2,2′-diamino-1,1′-binaphthyl, similar results were observed (entry 4). Nevertheless, employing a 1,2-cyclohexanediamine-based salen-Ti **III** led to an increased yield of 81% for compound **3a** (entry 5), whereas a bulky salen-Ti **IV** resulted in a maximum yield of 89% (entry 6).Furthermore, other solvents were explored and it was found that the reaction proceeded, albeit with lower yields compared to 1,4-dioxane (entries 7-11). Replacing Et$_3$N•HCl with 2,4,6-collidine•HCl led to a minimal amount of **3a** being obtained (entry 12). It is worth noting that the reaction still achieved a satisfactory yield when Zn dust was used as a reductant (entry 13). Attempting to reduce the Mn loading from 2 equiv to 0.5 equiv resulted in significantly lower

product yield (entry 14). In addition, lower yields were observed when reducing the amount of Et$_3$N•HCl (entries 15 and 16). It should be noted that the reaction did not proceed in the absence of Ti catalysts and reductants (entries 17 and 18).

### Substrate scope

Under optimized conditions, the present study investigated the compatibility of different 1,3-dienes with the salen-Ti **IV** catalyzed [2π + 2σ] cycloaddition reaction. The findings for substituted 1,3-dienes and BCB ketones are summarized in Fig. 2. The results indicate that a wide range of functionalized 1,3-dienes were tolerated, including aryl- and 1,1-disubstituted 1,3-dienes. This led to the production of a diverse series of 1,2,4-trisubstituted BCHs with comparable yields. Initially, the electronic effect of phenyl substituted 1,3-dienes was investigated. Surprisingly, phenyl derivatives with electron-neutral (-H), electron-donating (-Me, -OMe, -SMe or −NMe$_2$), and electron-withdrawing (-F, -Cl, -Br, -CF$_3$, -CN or -Ph) groups at the *para-*, *meta-*, or *ortho*-position were well tolerated (**3a-3m**). Additionally, naphthalene proved to be a competent partner, yielding the corresponding bicyclo[2.1.1]hexane in 61% yield (**3n**). Importantly, disubstituted phenyl 1,3-dienes with heteroatoms were all successfully accommodated (**3o,3p and 3r**), and the benzene ring containing active hydrogen also reacted smoothly, yielding product **3q** in a 66% yield. Notably, the 2,4,6-trifluoro-substituted phenyl was able to proceed smoothly and afforded product **3 s** in a 74% yield. Stilbenes bioisosteres with benzodioxole and benzo-dioxan groups were successfully synthesized (**3t** and **3 u**). Subsequently, a series of heteroaromatic substituted 1,3-dienes were screened under the salen-Ti **IV** catalyzed [2π + 2σ] cycloaddition conditions. Excitingly, substrates containing O, N, and S atoms were accommodated well, resulting in the desired products with excellent yields (**3v-3aa**). Various 1,1-disubstituted 1,3-dienes were also investigated, demonstrating that aryl (**4a-4e**), diphenyl (**4 f**), and alkyl (**4 g** and **4 h**) substituted 1,3-dienes were suitable coupling partners and proceeded smoothly. It is noteworthy that the elusive *trans*-configuration for the alkenes is observed in all of the aforementioned examples, including trisubstituted olefins, which were often challenging to acquire. 1,3-Cyclohexadiene was also well tolerated, producing tricyclo-decene **5a** in an excellent yield. The reaction was successfully applied to both 1,3-butadiene and 2,3-dimethylbuta-1,3-diene (**5b** and **5c**). 1-Alkyl substituted 1,3-dienes were converted to products **5d-5f** in moderate yields as a mixture of *E/Z*-isomers. In addition, 1,4-disubstituted 1,3-dienes were also tolerated, yielding products **5 g** and **5 h** in a 73% and 68% yields as a mixture of two isomers. The scope of BCB ketones were also briefly examined. The results indicate that piperonyl, naphthyl, and butyl were all tolerated, gave the corresponding products **6a-6c** in 53% to 97% yields. Furthermore, chlorophenyl substituted BCB ketones also reacted smoothly with 1-phenylbutadiene and butadiene (**6d** and **6e**).

Building on these success, we aimed to expand the scope of our study to include dienes with electron-withdrawing groups such as ester and amide. The outcome of our experiments is presented in Fig. 3. Notably, when using the standard conditions, we observed high yields in the reactions of a range of easily accessible 1,3-dienes with ester groups. Generally, the position of the substituent on the benzene ring did not have a significant impact on the reaction yields (**9a-9d**). Furthermore, we found that the presence of an aldehyde group was also compatible with the mild reaction conditions, resulting in a 73% yield of product **9e**. The investigation of amide groups demonstrated that both secondary amide (**9 f**) and tertiary amides (**9g-9i**) were suitable substrates, leading to the formation of the corresponding products in favorable yields. Additionally, we were pleased to observe that amide **7j** also reacted under the current conditions, resulting in a 74% yield of product **9j**. 1,3-Enynes are readily available organic compounds and their functionalization has emerged as a powerful method for synthesizing propargyl derivatives[64]. Excitingly, when selected phenyl

**Table 1 | Reaction optimization[a,b]**

| Entry | Variation from standard conditions | yield/% |
|---|---|---|
| 1 | none | 61 |
| 2 | CpTiCl$_3$ instead of Cp$_2$TiCl$_2$ | 19 |
| 3 | salen-Ti **I** instead of Cp$_2$TiCl$_2$ | 47 |
| 4 | salen-Ti **II** instead of Cp$_2$TiCl$_2$ | 58 |
| 5 | salen-Ti **III** instead of Cp$_2$TiCl$_2$ | 81 |
| 6 | salen-Ti **IV** instead of Cp$_2$TiCl$_2$ | 91(89)[c] |
| in entries 7–18, salen-Ti **IV** (5 mol%) was used | | |
| 7 | CH$_3$CN instead of 1,4-dioxane | 70 |
| 8 | DCM instead of 1,4-dioxane | 56 |
| 9 | EtOAc instead of 1,4-dioxane | 48 |
| 10 | THF instead of 1,4-dioxane | 83 |
| 11 | Et$_2$O instead of 1,4-dioxane | 39 |
| 12 | 2,4,6-collidine•HCl instead of Et$_3$N•HCl | 26 |
| 13 | Zn instead of Mn | 83 |
| 14 | Mn (50 mol%) | 44 |
| 15 | Et$_3$N•HCl (50 mol%) | 53 |
| 16 | no Et$_3$N•HCl | 17 |
| 17 | no Mn | 0 |
| 18 | no Ti-catalyst | 0 |

[a] Reaction scale: **1a** (0.2 mmol, 1equiv), **2a** (1.5 equiv). [b] Yields were determined by [1]H NMR spectroscopy vs. an internal standard (1,2,3-trimethoxybenzene) [c] isolated yield. CH$_3$CN Acetonitrile, DCM Dichloromethane, EtOAc Ethyl acetate, THF Tetrahydrofuran, Et$_2$O Diethyl ether.

substituted 1,3-enyne was used as the coupling partner, alkynyl substituted BCH scaffold product **10a** was isolated in 87% yield. To unfold the universality of the current reaction system to 1,3-enynes, we have extended the structures of several 1,3-enynes. The results indicate that substrates with -Me, -tBu or -F on the phenyl ring delivered the BCH products in high yields (**10b**-**10d**). Alkyl-substituted substrates could also be smoothly converted into the target product **10e** in 94% yield. In addition, a late-stage functionalization of drug derivatives was conducted. Various structurally complex 1,3-dienes derived from natural products and active drugs, including bezafibrate **11a**, febuxostat **11b**, and oxaprozin **11c**, were examined. All compounds proceeded well under optimal conditions, producing the corresponding products with comparable yields. These results highlight the robustness and utility of the current strategy in constructing various substituted bicyclo[2.1.1]hexane skeletons as structurally diverse stilbene bioisosteres.

## Synthetical application
To validate the practicality of the current methodology, a scale-up experiment and post-functionalization reactions were conducted (Fig. 4a). BCB ketone **1a** (1.35 g) was reacted 1,3-butadiene in the presence of 5.0 mol% of salen-Ti **IV**, resulting in the desired product **5b**

with a yield of 78% (1.31 g). In order to demonstrate the synthetic potential of this approach, several transformations of product **5b** were performed. For example, **5b** was subjected to epoxidation of the terminal olefin using *meta*-chloroperbenzoic acid (*m*-CPBA), yielding the epoxide product **12** with an 84% yield. The carbonyl group of **5b** could be efficiently reduced in the presence of NaBH$_4$, affording the corresponding product **13** with a 98% yield (d.r. = 2:1). Additionally, a Wacker-type olefin oxidation of **5b** was carried out under the same conditions as previously reported in the literature[65], resulting in a 68% yield of the ketone **14**. Methylthioether functional groups are well-known for their unique biological activity and significant value in the field of medicine[66]. Therefore, we successfully introduced the methylthio group into the BCH skeleton **15** by photocatalytic triplet-triplet energy transfer pathway[67]. Intriguingly, upon irradiation of the trans-configured alkene **3r** using 400 nm LEDs for a duration of 12 h, the *E/Z* ratio transformed to 1.4:1, and product **3r'** was separated in a 41% yield as a *cis*-configuration alkene[68]. In addition to the aforementioned results, successful conversion from amide to ketone was also achieved (Fig. 4c), resulting in the corresponding product **9k** with a yield of 95%. These successful application examples further demonstrate the practicality and applicability of the Ti-catalyzed [2σ + 2π] cycloaddition of BCB ketones with 1,3-dienes.

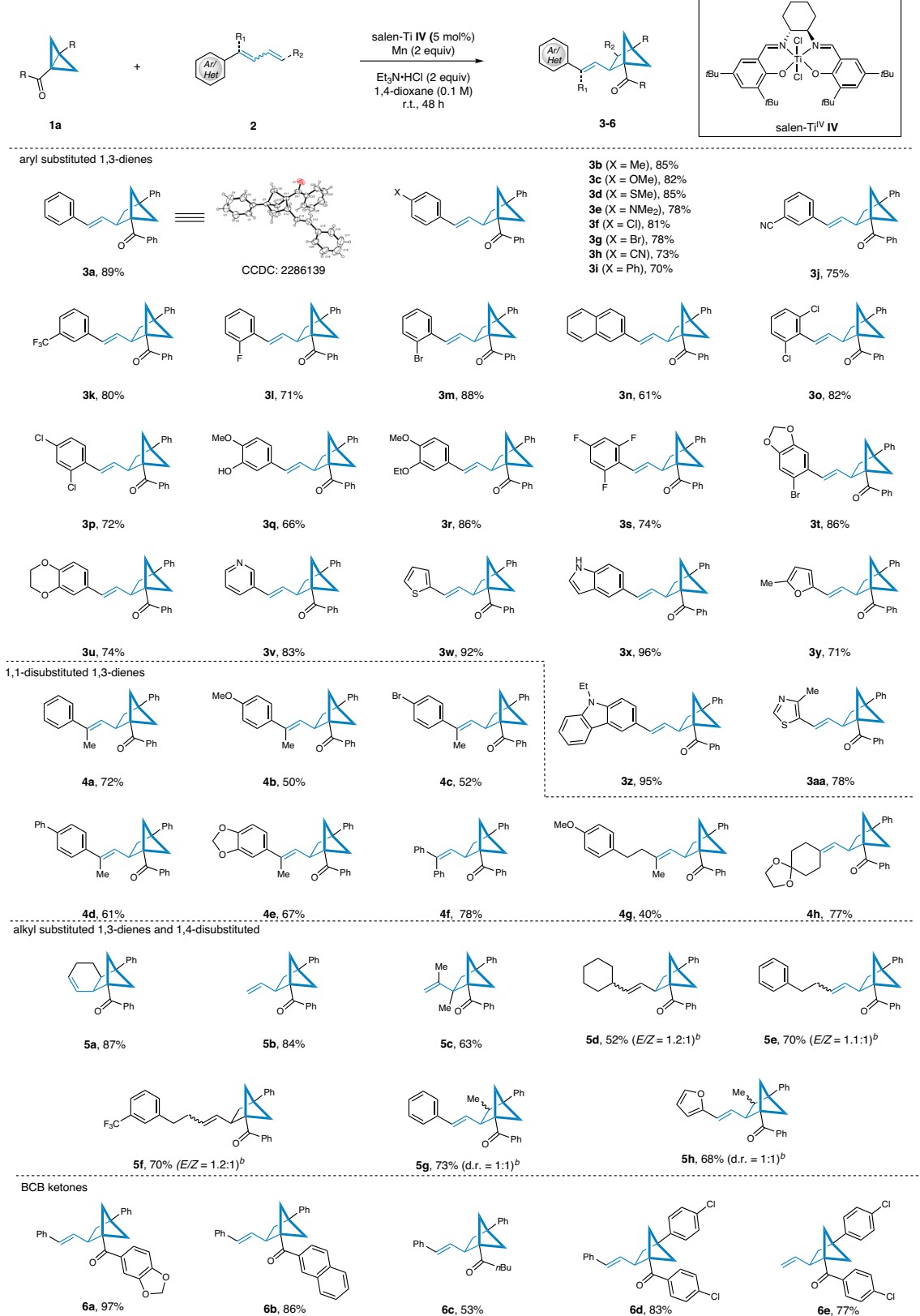

**Fig. 2 | Substrate scope towards1,3-dienes and BCB ketones.** Reaction conditions: **a 1** (0.2 mmol, 1 equiv), **2** (1.5 equiv), salen-Ti **IV** (5 mol%), Mn (2 equiv) and Et₃N•HCl (2 equiv) in 1,4-dioxane (0.1 M) stirred for 48 h under N₂ conditions; **b** E/Z ratio and d.r. were determined by ¹H NMR analysis of crude products. **c** Isolated yield.

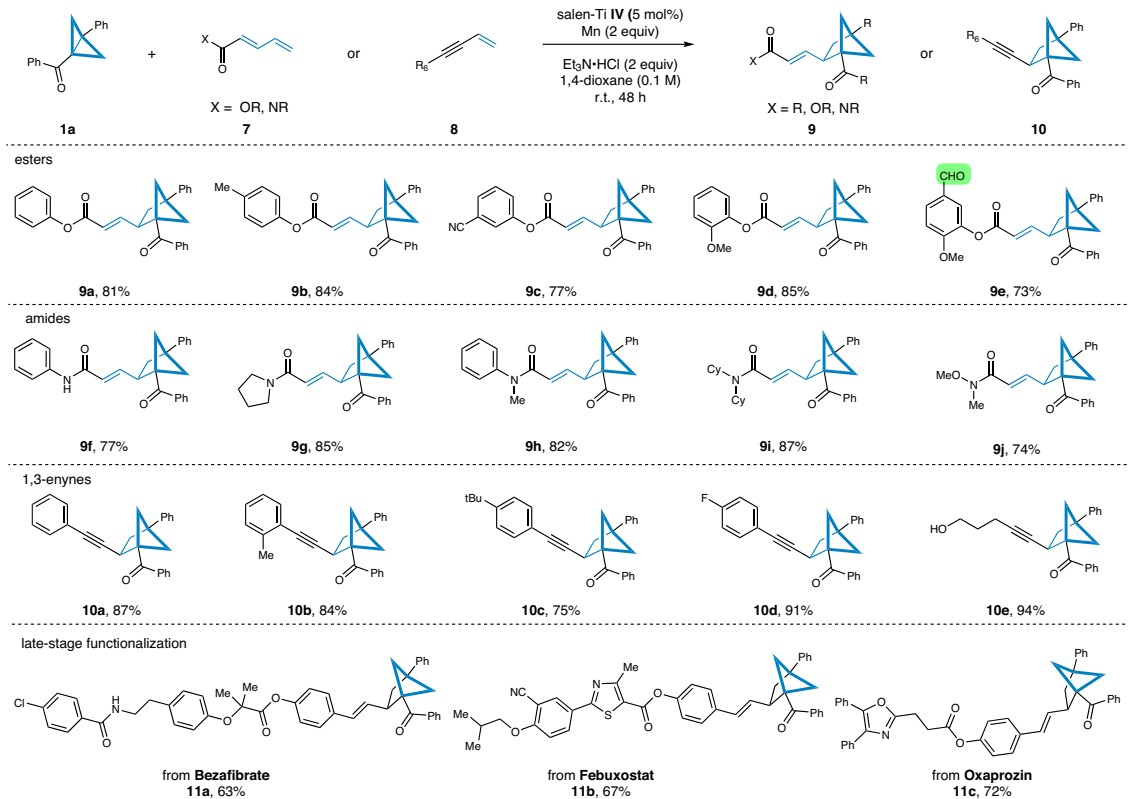

**Fig. 3 | Substrate scope towards 1,3-diene substituted with electron-withdrawing groups.** Reaction conditions: **a 1a** (0.2 mmol, 1 equiv), **7** or **8** (1.5 equiv), salen-Ti **IV** (5 mol%), Mn (2 equiv) and Et₃N·HCl (2 equiv) in 1,4-dioxane (0.1 M) stirred for 48 h under N₂ conditions. **b** Isolated yield.

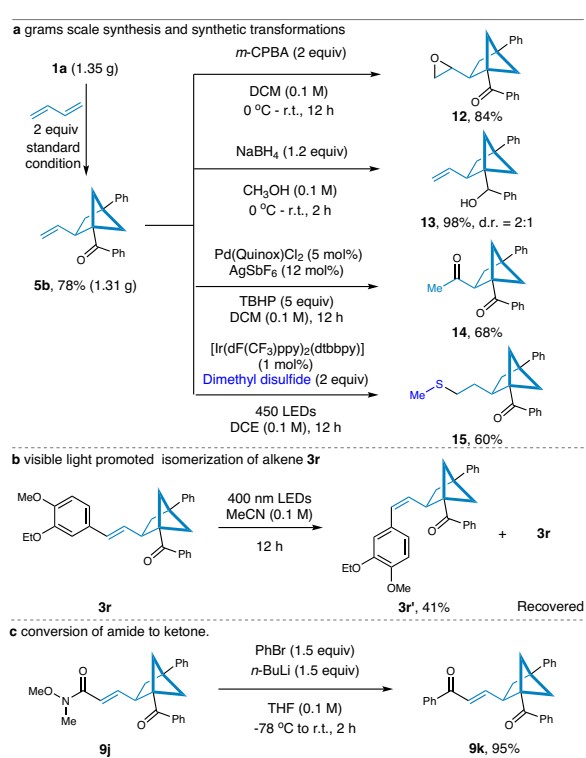

**Fig. 4 | Synthetic applications. a** Grams scale synthesis and synthetic transformations. **b** Conversion of *trans*-alkene to *cis*-alkene under 400 nm LEDs irradiation. **c** Conversion of amide **9j** to ketone **9k**. *m*-CPBA *meta*-chloroperbenzoic acid, *n*-BuLi *n*-Butyllithium, DCE 1,2-Dichloroethane.

## Mechanistic studies

In addition, a radical trapping experiment was conducted to elucidate the reaction mechanism. Upon addition of the scavenger TEMPO, the formation of the desired product **3a** was completely inhibited, and instead, the radical-trapping adduct **16** was detected by high-resolution mass spectrometry (HRMS) (Fig. 5a, for more details see the Supplementary Fig. 6). This finding provides further support for the involvement of radical intermediates in the salen-Ti **IV** catalyzed [2π + 2σ] cycloaddition reaction.

DFT calculations were also performed to gain more insights into the origin of *E/Z*-selectivity and regioselectivity ([4 + 2] vs [2 + 2]) at the ωB97XD/def2-TZVPP/SMD (solvent = 1,4-dioxane)//ωB97XD/def2-SVP level of theory. We set out to explore the *trans*-selectivity based on two model allyl radicals **17** and **18**. With a free energy barrier of only 11.4 kcal/mol, the *cis*-configuration **17′** would convert predominantly to the thermodynamically more favorable *trans*-configuration **17**, which was 3.8 kcal/mol more stable (Fig. 5b). Therefore, a rapid *cis*- to *trans*-isomerization would occur during the formation of allyl radicals **D** despite the initial presence of both *E/Z* isomers of 1-phenyl-1,3-butadiene **2a**, leading to the exclusive formation of *trans*-configurations. Similarly, the *trans*-1-cyclohexyl-1,3-butadiene **18** was also calculated to be more stable, albeit to a smaller extent of 0.8 kcal/mol compared to **18′** (Fig. 5c), in consistent with experimental observations of products **5d** (*E/Z* = 1.2:1), **5e** (*E/Z* = 1.1:1) and **5 f** (*E/Z* = 1.2:1).

The subsequent radical-enolate addition could potentially result in the formation of two regioisomeric products (Fig. 5d). In alignment with the experimental observations, our calculations on this step revealed a remarkable preference for the pathway leading to the [2 + 2] cycloaddition products **E**. This pathway, with a free energy barrier of 16.3 kcal/mol, was significantly more accessible compared to the [4 + 2] cycloaddition, which has a much higher energy barrier

**Fig. 5 | Mechanistic studies. a** Radical trapping experiment in the presence of 2,2,6,6-Tetramethylpiperidinooxy (TEMPO). **b** DFT calculations for the *E/Z*-selectivity of 1-phenyl-1,3-butadiene. **c** DFT calculations for the *E/Z*-selectivity of 1-cyclohexyl-1,3-butadiene. **d** DFT calculations for the regioselectivity of [4 + 2] vs [2 + 2]. Transition states are illustrated using CYLview20[69].

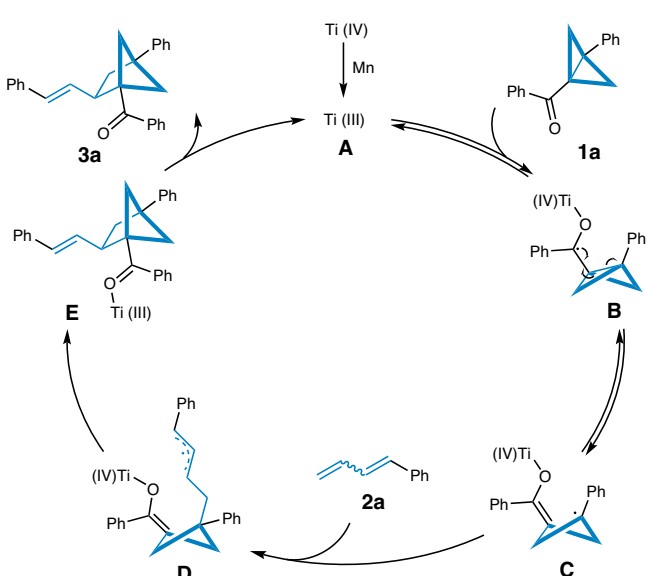

**Fig. 6 | Proposed catalytic mechanism.** The corresponding mechanism for the selected model reaction between BCB **1a** and 1-phenyl-1,3-butadiene **2a**.

of 38.9 kcal/mol. Additionally, the [2 + 2] cycloaddition product **E** was thermodynamically more stable than its [4 + 2] product **E'** counterparts by a substantial margin of 35.2 kcal/mol. This stability for both transition state and product was presumably due to the dihedral angle difference observed in double bonds: 171.9° for [2 + 2] and 121.3° for [4 + 2], respectively, suggesting a distorted alkene structure (atoms labeled in pink). In order to further examine the influence of different titanium catalysts on the energy profile and regioselectivity, calculations were also conducted using $Cp_2Ti^{IV}Cl$ as the [Ti] component, and even with [Ti] = H by replacing the [Ti] moiety with a hypothetical hydrogen atom. Interestingly, computational results showed that the energy profiles and regioselectivity towards both [4 + 2] and [2 + 2] cycloaddition pathways remained largely unchanged, regardless of the nature of titanium catalyst used, indicating that the inherent regioselectivity of the reaction is primarily governed by the substrate structures. This conclusion underscores the robustness of the [2 + 2] cycloaddition preference under a variety of catalytic scenarios.

## Proposed mechanism

Base on this result and previous reports[62,63], A plausible reaction pathway is proposed in Fig. 6. Initially, by addition of an excess amount of Mn powder, the $Ti^{IV}$ precursor can be reduced to obtain the species

Ti$^{III}$. A reversible single-electron transfer (SET) between the Ti$^{III}$ species **A** and the BCB ketone **1a** results in the formation of a radical intermediate **B**. This intermediate undergoes reversible intramolecular ring opening to generate the enolate radical species **C**. Subsequently, intermolecular coupling with a mixture of *E/Z* isomers of 1-phenyl-1,3-butadiene **2a** produces allyl radicals **D**. The allyl radicals then add to the Ti$^{IV}$–enolate moiety, leading to the formation of new ketyl allyl radicals **E**. Finally, the desired product **3a** is obtained through an intramolecular SET of **E**, and the Ti$^{III}$ catalyst is regenerated, completing the catalytic cycle.

## Discussion

In summary, we have developed a reliable synthetic strategy for the Ti-catalyzed [2σ + 2π] cycloaddition of bicyclo[1.1.0]-butanes with 1,3-dienes or feedstock butadiene. The mildness of this protocol allows for the introduction of various substituents, including aryl, alkyl, 1,1- and 1,4-disubstituted 1,3-dienes, 1,3-enynes, ester and amide. The synthetic significance of this versatile approach is demonstrated through late-stage modification of pharmaceuticals, gram-scale synthesis, and subsequent conversion of the bicyclic hexane (BCH) skeletons. This method enables the efficient synthesis of 1,2,4-trisubstituted bicyclo[2.1.1]hexanes as bioisosteres of stilbenes.

## Methods

### General procedure

In a flame-dried 10 mL reaction tube equipped with a magnetic stirrer bar was charged sequentially with salen-Ti **IV** (0.05 equiv), Mn (2.0 equiv), Et$_3$N•HCl (2.0 equiv). The Schlenk flask was transferred to an argon-filled glovebox and followed by the addition of 1,4-dioxane (2 mL), and the mixture was stirred at room temperature for 30 min in argon-filled glovebox. Then, to the resulting mixture were added 1,3-dienes (0.3 mmol, 1.5 equiv) and BCB ketones (0.2 mmol, 1.0 equiv). The resulting mixture was removed out the glovebox and stirred at room temperature for 48 h. After completion of the reaction, the mixture was filtered by silicone gasket. The solvent was evaporated in vacuo and the crude material was purified by flash column chromatography to furnish the desired product.

## Data availability

The authors declare that the data supporting the findings of this study are available within the article and Supplementary Information file, or from the corresponding author upon request. The data for the coordinates of the optimized structures are present in a source data file. Crystallographic data for the structures reported in this article have been deposited at the Cambridge Crystallographic Data Center, under deposition numbers CCDC 2286139. Copies of the data can be obtained free of charge via https://www.ccdc.cam.ac.uk/structures/. Source data are provided with this paper.

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

## Acknowledgements

This project is supported by the National Natural Science Foundation of China (22171036), the Natural Science Foundation of Henan Province (Nos. 232300421126), Open Research Fund of School of Chemistry and Chemical Engineering, Henan Normal University (2020YB03).

## Author contributions

L.S. conceived the study. Y.L. discovered and conducted experiments. Z.W. and H.Y. synthesized the BCB ketones and dienes. J.-R.S. conducted

DFT calculations. L.S. and Y.L. proposed the mechanism. Y.L., E.H. and L.S. wrote the manuscript.

## Competing interests

The authors declare no competing interests.
