## [Peer Review File · Nature Communications]

REVIEWER COMMENTS

Reviewer #1 (Remarks to the Author):

The manuscript from Prof. Shi's group describes a titanium-catalyzed formal cycloaddition between bicyclo[1.1.0]butanes (BCBs) and 1,3-dienes to access a variety of alkenyl substituted bicyclo[2.1.1]hexanes (BCHs) with excellent regio- and trans-selectivity. Mechanistically, the reaction involves a radical redox relay strategy well illustrated in the [3+2] cycloaddition of cyclopropyl ketones and alkenes (Lin, S et al. *J. Am. Chem. Soc.* 140, 3514-3517 (2018)). This developed method presented a new approach to construct structurally intriguing multi-substituted bicyclo[2.1.1]hexanes as stilbene bioisosteres with potential medicinal application. Impressively, the reaction could extend to dienes such as cyclohexadiene or butadienes. Moreover, the existing olefin/alkyne unit would be ready to undergo further transformations, generating other derivatives. The utilities of this transformation have been well demonstrated by late-stage modification of drug derivatives. Based on these aspects, this referee would recommend this manuscript to be published on *Nature Communications* after addressing the following issues.

- 1) Does the author examine the enantioselectivity of this transformation via a chiral titanium catalyst? The asymmetric synthesis of BCH remains a formidable challenge and would be highly significant.
- 2) In the manuscript, the reaction underwent a similar mechanism to SmI₂-mediated formal cycloaddition as shown in Procter's work (*Nat. Chem.* 15, 535-541 (2023)) where 3-alkyl substituted BCBs and electron-deficient alkenes could work very well. However, in this manuscript, only 3-aryl substituted BCB reaction partners were displayed. Does the reaction with 3-alkyl substituted BCB proceed? Could the reaction be extended to dienes with electron-withdrawing groups such as ketone, ester, nitrile, and amide? Could the author give any explanation on why Ti- and Sm-system have different substrate tolerance?
- 3) In line 103, it is suggested that the -Cl or -Ph should be electron- withdrawing group and included in electron-withdrawing (-F, -Cl, -Br, -CF₃, -CN or -Ph) groups.
- 4) Why did this reaction give trans-configuration alkene exclusively with a mixture of E/Z isomers of dienes? Does the author observe any formal [4+2] cycloaddition product?
- 5) The reaction with 1,3-enynes could afford alkynyl-substituted BCH 6, which is useful in the synthesis of cis-stilbene bioisosteres via Lindlar hydrogenation. It is suggested that the hydrogenation of alkyne to Z-alkene should be included in the synthetic application to enrich the utility of this protocol.
- 6) Some suggestions in the Supporting Information section: the HR-MS data of 1a and 1b should be supplemented; some detailed information about titanium catalysts would be better included; ¹H NMR spectrum of compound 4g in Page 71 poses some peculiarity.

Minor typos and errors in the manuscript and supporting information section are listed below:

- 1) In line 11 and 60, bicyclo[2.1.1]hexane (BCH) was incorrectly abbreviated as BHC. It would be better to give the complete name with its abbreviation like "bicyclo[2.1.1]hexane (BCH)" for its first appearance in line 11.
- 2) In lines 62, 146, and 147, "structral divers" should be corrected as "structurally diverse".
- 3) In line 109, "3s" should be changed to "3q"; in line 110, "3t" should be changed to "3s"; in line 112, "3u and 3v" should be changed to "3t and 3u".
- 4) The alkyne structural formula of compound 6 in Table 2 should be linear due to sp hybridization like 6a-6e.
- 5) In Table 2, "1,3-denyne" should be "1,3-enynes".
- 6) Line 148, "1a" should be bold as "1a"; line 175, "A" should be bold as "A".
- 7) In Supporting Information, for compound 3a, "13C NMR-DMPT" should be "13C NMR-DEPT".

Reviewer #2 (Remarks to the Author):

In this manuscript, the authors have described their discovery of a Ti-catalyzed radical cycloaddition reaction that can be used for synthesis of a variety of stilbene-like bicyclic[2.1.1]hexanes. As the BCHs are bioisosteres of benzene ring and have received significant attention in recent drug R&D, new methods for rapid synthesis of related molecules are highly attractive. This work provides a novel, broadly compatible and highly effective approach for the target products. The data are solid and the presentation is clear and concise. The work may be publishable in Nat. Commun. after the following points have been appropriately addressed.

- (1) Can internal diene be used for the reaction? Examples should be provided.
- (2) Do the authors observe any 1,4-cycloaddition products other than the 1,2-cycloaddition major products?
- (3) Substituents other than an aryl group at the bridgehead of [1.1.0] reactant should be tested.

(4) Since it is a very active direction in using bicyclo[1.1.0]butane for transformations, the authors should carefully check cite more references, such as Angew. Chem.Int. Ed.2023,e202310066.

Reviewer #3 (Remarks to the Author):

The authors utilize titanium catalysis to couple bicyclobutanes with dienes (or yneenes). Interesting product motifs result, bioisosteres of stilbenes/styrenes. The remaining double/triple bond can be readily functionalized, or left intact as an important structural unit. The authors provide a carefully written manuscript containing all the sections that would be expected. However, the mechanistic part could be improved (additional insightful experiments; DFT calculations), but this would nice but not crucial for publication.

I enjoyed reading this manuscript and it seems suitable for NatCommun. No mistakes could be found. Very minor things might be changed though:

1) Even if this manuscript does not focus on photocatalysis, the following review on closely related strain release transformations would be insightful and should be cited:

Strain-Release Photocatalysis,

J. Am. Chem. Soc. 2023, 145, 20716

2) Italicize trans, ortho, meta, ...

3) Check use of capital letters for consistency (for example key words).

Reviewer 1: Comments: The manuscript from Prof. Shi's group describes a titanium-catalyzed formal cycloaddition between bicyclo[1.1.0]butanes (BCBs) and 1,3-dienes to access a variety of alkenyl substituted bicyclo[2.1.1]hexanes (BCHs) with excellent regio- and trans-selectivity. Mechanistically, the reaction involves a radical redox relay strategy well illustrated in the [3+2] cycloaddition of cyclopropyl ketones and alkenes (Lin, S et al. *J. Am. Chem. Soc.* **140**, 3514-3517 (2018)). This developed method presented a new approach to construct structurally intriguing multi-substituted bicyclo[2.1.1]hexanes as stilbene bioisosteres with potential medicinal application. Impressively, the reaction could extend to dienes such as cyclohexadiene or butadienes. Moreover, the existing olefin/alkyne unit would be ready to undergo further transformations, generating other derivatives. The utilities of this transformation have been well demonstrated by late-stage modification of drug derivatives. Based on these aspects, this referee would recommend this manuscript to be published on *Nature Communications* after addressing the following issues.

Many thanks for your kind comments, following is the response to your suggestions and questions.

Question 1: Does the author examine the enantioselectivity of this transformation via a chiral titanium catalyst? The asymmetric synthesis of BCH remains a formidable challenge and would be highly significant.

Response: Thank you for your valuable suggestion. We greatly acknowledge and share your perspective on the significant and challenging nature of asymmetric synthesis of BCH. In our research, we employed BCB ketone **1a** and 1-phenyl-1,3-butadiene **2a** as coupling partners. Initially, we conducted optimizations on the salen-Ti catalyst, which resulted in a mere 5% ee when utilizing salen Ti I. However, when styrene were used, we were able to achieve 27% ee by employing salen-Ti I. Drawing from these findings and your insightful suggestions, our research group is actively investigating the salen-Ti catalytic system for the synthesis of chiral bioisosteres. The following section outlines the experimental results obtained from our investigations.

Question 2: In the manuscript, the reaction underwent a similar mechanism to SmI₂-mediated formal cycloaddition as shown in Procter's work (Nat. Chem. **15**, 535-541 (2023)) where 3-alkyl substituted BCBs and electron-deficient alkenes could work very well. However, in this manuscript, only 3-aryl substituted BCB reaction partners were displayed. Does the reaction with 3-alkyl substituted BCB proceed? Could the reaction be extended to dienes with electron-withdrawing groups such as ketone, ester, nitrile, and amide? Could the author give any explanation on why Ti- and Sm-system have different substrate tolerance?

Response: Thanks for your kind suggestion. We have examined Procter's work and, to broaden our study, we have included a range of different substrates for comparison. The following are new examples of BCB ketones and 1,3-dienes with electron-withdrawing groups. These results have been added into revised manuscript, page 8 (figures 2), page 9 (figures 3), and page 11 (figures 4).

1) We examined the reactivity of 3-phenyl substituted BCB ketones, namely piperonyl **6a**, naphthyl **6b**, and butyl **6c**. All three substrates exhibited smooth reactivity.

2) We next tested 1,3-dienes with electron-withdrawing groups such as ester, amide and ketone. The results showed that the reaction could proceed smoothly when using esters

(**9a-9e**) and amides (**9f-9j**). We found that the current conditions are not optimal for ketone substrates. Our laboratory is actively exploring the optimal conditions for such substrates and will report it in due course. However, we obtained ketone product **9k** through subsequent conversion of amide product **9j**. We did not attempt nitrile substituted 1,3-diene because the raw materials were not easily obtainable. Although there is a literature describing the synthesis method (J. Am. Chem. Soc. 1949, 71, 1055), the raw material NaCN required for the experiment is a controlled drug and is not allowed to be used in the laboratory.

3) Furthermore, we observed that the 3-methyl-substituted BCB ketone did not undergo the anticipated reaction and was fully recovered. Our speculation is that this result may be attributed to the comparatively lower reduction capability of the salen-Ti catalyst in comparison to SmI_2 .

Question 3: In line 103, it is suggested that the -Cl or -Ph should be electron- withdrawing group and included in electron-withdrawing (-F, -Cl, -Br, -CF₃, -CN or -Ph) groups.

Response: Thanks for suggestion. It has been modified in the revised manuscript. The following is the modified content.

Surprisingly, phenyl derivatives with electron-neutral (-H), electron-donating (-Me, -OMe, -

SMe or $-NMe_2$), and electron-withdrawing ($-F$, $-Cl$, $-Br$, $-CF_3$, $-CN$ or $-Ph$) groups at the *para*-, *meta*-, or *ortho*-position were well tolerated. Page 6.

Question 4: Why did this reaction give *trans*-configuration alkene exclusively with a mixture of *E/Z* isomers of dienes? Does the author observe any formal [4+2] cycloaddition product?

Response: Many thanks for your question. To clarify this reason, we conducted DFT calculations. DFT calculations showed that both (*Z/E*)-1-phenyl-1,3-butadiene would convert predominantly to the thermodynamically more favorable *E*-configurations allyl radical, which was 3.8 kcal/mol more stable. Therefore, a rapid *cis*- to *trans*- isomerization would occur during the formation of allyl radicals D despite the initial presence of both *E/Z* isomers of 1-phenyl-1,3-butadiene **2a**, leading to the exclusive formation of *trans*-configurations product. In this reaction system, we did not observe any products of 4+2 cycloaddition. DFT calculation discovery that the [2+2] cycloaddition product was thermodynamically more stable than its [4+2] counterparts by a substantial margin of 35.2 kcal/mol. Detailed discussions please found in manuscript page 12 (figures 5b, 5c and 5d).

Question 5: The reaction with 1,3-enynes could afford alkynyl-substituted BCH **6**, which is

useful in the synthesis of *cis*-stilbene bioisosteres via Lindlar hydrogenation. It is suggested that the hydrogenation of alkyne to *Z*-alkene should be included in the synthetic application to enrich the utility of this protocol.

Response: We greatly appreciate your suggestion. We conducted Lindlar hydrogenation on product **10a**; however, we were unable to obtain the desired *cis*-product. This outcome may be attributed to steric hindrance. Considering the significance of *cis*-stilbenes as drug molecules, we aimed to explore alternative methods to obtain *cis*-products. Through our investigation, we discovered that when the *trans*-type alkene **3r** was exposed to 400 nm LEDs for a duration of 12 hours, the *E/Z* ratio shifted to 1.4:1, resulting in the formation of a separated product **3r'**. This product was obtained with a yield of 41% as a *cis*-type alkene. The experimental results are as follows. These results have been incorporated into the revised manuscript on page 11, specifically in Figure 4.

Question 6: Some suggestions in the Supporting Information section: the HR-MS data of **1a** and **1b** should be supplemented; some detailed information about titanium catalysts would be better included; ¹H NMR spectrum of compound **4g** in Page 71 poses some peculiarity.

Response: Many thanks for your suggestions. We carefully checked our Supporting Information and provided a revised ESI.

1) We have added HR-MS for **1a** and **1b** in page 5. Following are the results of the modifications.

1a, HRMS: calculated for C₁₇H₁₅O [M+H]⁺ 235.1117; found 235.1114. Page 5.

1b, HRMS: calculated for C₁₇H₁₃Cl₂O [M+H]⁺ 303.0338; found 303.0351. Page 5.

2) We have added the synthesis method of salen-Ti catalyst as follows. These parts have been added into revised Supporting Information in page 4

To a heat-dried Schlenk flask were added the salen ligand (5.0 mmol, 1.0 eq) and freshly distilled THF. The resulting yellow solution was cooled to -78 °C under N₂. Then TiCl₄ solution (1.0 M in toluene, 1.0 eq) was added dropwise into the above solution at the same temperature. The red suspension was then allowed to warm to r.t. and heated under reflux for 2 h. After the reaction was cooled to room temperature, the red solid was obtained by recrystallization with DCM and n-hexane. Then the red solid was washed with diethyl ether and n-hexane to afford salen-Ti complex.

3) We have carefully examined the spectrum on page 71 of the old version and have been changed from ¹H NMR to 1D NOE (page 91).

Minor typos and errors in the manuscript and supporting information section are listed below:

Question 1: In line 11 and 60, bicyclo[2.1.1]hexane (BCH) was incorrectly abbreviated as BHC. It would be better to give the complete name with its abbreviation like “bicyclo[2.1.1]hexane (BCH)” for its first appearance in line 11.

Response: Thanks for your careful reading. It has been corrected in the manuscript. We also provided the full name for its first appearance (page 1).

Question 2: In lines 62, 146, and 147, “strucrtal divers” should be corrected as “structurally diverse”.

Response: Thanks for your careful reading. We carefully examined the spelling of words, and now have corrected the “strucrtal divers” to “structurally diverse”. Pages 4 and 10.

Question 3: In line 109, “3s” should be changed to “3q”; in line 110, “3t” should be changed to “3s”; in line 112, “3u and 3v” should be changed to “3t and 3u”.

Response: Thanks for your careful reading. We carefully examined the number of compounds, and now have corrected in the manuscript (pages 6 and 4). Specifically, **3s** has been changed to **3q**, **3t** to **3s**, **3u** to **3t** and **3v** to **3u**. Pages 6 and 7.

Question 4: The alkyne structural formula of compound **6** in Table 2 should be linear due to sp hybridization like **6a-6e**.

Response: Thanks for your careful reading. It has been corrected in the manuscript (page 9).

Question 5: In Table 2, “1,3-denyne” should be “1,3-enynes”.

Response: Many thanks for your careful reading. The spelling has been changed in revised manuscript (page 9). Specifically, “1,3-denyne” has been changed to “1,3-enynes”.

Question 6: Line 148, “1a” should be bold as “**1a**”; line 175, “A” should be bold as “**A**”.

Response: Many thanks for your careful reading. The corresponding font has been bolded in the manuscript (pages 10 and 13).

Question 7: In Supporting Information, for compound **3a**, “¹³C NMR-DMPT” should be “¹³C NMR-DEPT”

Response: Thanks for your careful reading. It has been corrected in supporting information (pages 30 and 53). Specifically, “¹³C NMR-DMPT” has been changed to “¹³C NMR-DEPT”.

Reviewer 2: Comments: In this manuscript, the authors have described their discovery of a Ti-catalyzed radical cycloaddition reaction that can be used for synthesis of a variety of stilbene-like bicyclic[2.1.1]hexanes. As the BCHs are bioisosteres of benzene ring and have received significant attention in recent drug R&D, new methods for rapid synthesis of related molecules are highly attractive. This work provides a novel, broadly compatible and highly effective approach for the target products. The data are solid and the presentation is clear and concise. The work may be publishable in Nat. Commun. after the following points have been appropriately addressed.

Many thanks for your positive comments, following is the response to your kind suggestions and questions.

Question 1: Can internal diene be used for the reaction? Examples should be provided.

Response: We appreciate your valuable suggestion. In our study, we utilized BCB **1a** in combination with a 1,4-disubstituted 1,3-diene **2** as coupling partners, following standard conditions. During the preliminary screening of internal dienes, we observed that the reaction proceeded smoothly only when methyl was utilized. However, when isopropyl was used, the desired product could not be obtained. This discrepancy may be attributed to the greater steric hindrance associated with isopropyl, in comparison to methyl. The following are examples of internal dienes. These results have been added into revised manuscript, page 8 (figure 2).

Question 2: Do the authors observe any 1,4-cycloaddition products other than the 1,2-cycloaddition major products?

Response: We greatly appreciate your question. In this reaction system, we did not

observe any products of 4+2 cycloaddition. In order to reveal this reason, we conducted DFT calculations. DFT calculations showed that the [2+2] cycloaddition product was thermodynamically more stable than its [4+2] counterparts by a substantial margin of 35.2 kcal/mol. Following are the calculation results. This part has been added into revised manuscript (page 12, figure 5c).

Question 3: Substituents other than an aryl group at the bridgehead of [1.1.0] reactant should be tested.

Response: Thanks for your kind suggestion. To broaden our study, we have included a range of different substrates for testing. We examined the reactivity of 3-phenyl substituted BCB ketones, namely piperonyl **6a**, naphthyl **6b**, and butyl **6c**. All three substrates exhibited smooth reactivity. These results have been added into revised manuscript (page 8, figure 2).

Question 4: Since it is a very active direction in using bicyclo[1.1.0]butane for transformations, the authors should carefully check cite more references, such as Angew.

Chem.Int. Ed. 2023, e202310066.

Response: Many thanks for your kind suggestion. We have thoroughly reviewed this article and have found it to be an excellent contribution in synthesizing BCH skeletons. We are delighted to cite this article as reference 59 on page 19.

Reviewer 3: Comments: The authors utilize titanium catalysis to couple bicyclobutanes with dienes (or yneenes). Interesting product motifs result, bioisosteres of stilbenes/styrenes. The remaining double/triple bond can be readily functionalized, or left intact as an important structural unit. The authors provide a carefully written manuscript containing all the sections that would be expected. However, the mechanistic part could be improved (additional insightful experiments; DFT calculations), but this would be nice but not crucial for publication. I enjoyed reading this manuscript and it seems suitable for NatCommun. No mistakes could be found. Very minor things might be changed though:

Many thanks for your positive comments. Following is the response to your suggestions and questions.

Given the consideration of *cis* and *trans* configurations of alkenes and the potential [4+2] cycloaddition pathways, we completely agree with your suggestion to perform DFT calculations. In summary, the DFT calculations demonstrated that, in the case of (*Z/E*)-1-phenyl-1,3-butadiene, the conversion primarily favored the formation of the thermodynamically more stable *E*-configurations allyl radical. This indicates that a rapid *cis*-to-*trans* isomerization occurs during the generation of allyl radicals **D**, resulting in the exclusive production of *trans*-configurations products, despite the presence of both *E* and *Z* isomers initially. Additionally, the DFT calculations revealed that the [2+2] cycloaddition product exhibited significantly greater thermodynamic stability compared to its [4+2] counterparts, with a substantial margin of 35.2 kcal/mol. For a more comprehensive discussion on these findings, please refer to the manuscript on page 12, accompanied by figures 5b, 5c, and 5d.

Question 1: Even if this manuscript does not focus on photocatalysis, the following review on closely related strain release transformations would be insightful and should be cited: Strain-Release Photocatalysis, *J. Am. Chem. Soc.* 2023, 145, 20716.

Response: Many thanks for your kind suggestion. We carefully read this article and found that it is an excellent review. We are happy to cite this article as reference 51 in page 18.

Question 2: Italicize trans, ortho, meta, ...

Response: Many thanks for you kind reminder. We carefully checked the English writing and provided a revised manuscript.

Question 3: Check use of capital letters for consistency (for example key words).

Response: We greatly appreciate your suggestion. We carefully checked the key words and full text. The first letters have already been capitalized

REVIEWERS' COMMENTS

Reviewer #1 (Remarks to the Author):

The authors have already addressed all the concerns raised by all referees, and the manuscript is suitable for publication on Nature Communications.

Reviewer #3 (Remarks to the Author):

The authors carefully revised the manuscript. In regard to my comments, I am especially happy that they conducted the DFT investigation I suggested. But I am also grateful for reviewer 1 for pointing out several little mistakes, they apparently read the paper more carefully than me. Only two comments remain:

Figure 1: "trans" should be italicized.

Throughout the manuscript, "regio" must not be italicized.

The manuscript can get accepted for publication!

Reviewer 1: Comments: The authors have already addressed all the concerns raised by all referees, and the manuscript is suitable for publication on Nature Communications.

We sincerely appreciate this reviewer for his/her favorable comments on our manuscript, and at the same time, we would like to acknowledge this reviewer for his/her contribution to improve the quality of our manuscript.

Reviewer 3: Comments: The authors carefully revised the manuscript. In regard to my comments, I am especially happy that they conducted the DFT investigation I suggested. But I am also grateful for reviewer 1 for pointing out several little mistakes, they apparently read the paper more carefully than me. Only two comments remain: Figure 1: "trans" should be italicized. Throughout the manuscript, "regio" must not be italicized. The manuscript can get accepted for publication!

We sincerely appreciate this reviewer for his/her favorable comments on our manuscript, and at the same time, we would like to acknowledge this reviewer for his/her contribution to improve the quality of our manuscript. Following is the response to your suggestions and questions.

Question 1: Figure 1: "trans" should be italicized.

Response: Thanks for your careful review. We carefully examined the format of words, and now have corrected the "trans" to "*trans*". (Figure 1)

Question 2: Throughout the manuscript, "regio" must not be italicized.

Response: Many thanks for your careful reading. The italic format of "region" has been corrected in revised manuscript. (Pages 1 and 2)